

# Measurement, growth types and shrinkage
# of newly formed aerosol particles at an urban research platform

Imre Salma[1], Zoltán Németh[1], Tamás Weidinger[2], Boldizsár Kovács[3], Gergely Kristóf [3]

[1]Institute of Chemistry, Eötvös University, H-1518 Budapest, P.O. Box 32, Hungary
[2]Department of Meteorology, Eötvös University, H-1518 Budapest, P.O. Box 32, Hungary
[3]Department of Fluid Mechanics, Budapest University of Technology and Economics, H-1111 Budapest, Bertalan L. u. 4–6., Hungary

*Correspondence to*: Imre Salma (salma@chem.elte.hu)

**Abstract.** Budapest platform for Aerosol Research and Training (BpART) was created for advancing long-term on-line
atmospheric measurements and intensive aerosol sample collection campaigns in Budapest. A joint study including
atmospheric chemistry/physics, meteorology and fluid dynamics on several-year long data sets obtained at the platform
confirmed that the location represents a well-mixed, average atmospheric environment for the city centre. The air streamlines
indicated that the host and neighbouring buildings together with the natural orography play an important role in the near field
dispersion processes. Details and features of the air flow structure were derived, and they can be readily utilised for further
interpretations. An experimental method to determine particle diffusion losses in the DMPS system of the BpART facility was
proposed. It is based on CPC/CPC and DMPS/CPC comparisons. Growth types of nucleated particles observed in 4 years of
measurements were presented and discussed for cities in focus. Arch-shape size distribution surface plots consisting of a
growth phase followed by a shrinkage phase were characterised separately since they supply information on nucleated
particles. They were observed in 4.5% of quantifiable nucleation events. The shrinkage phase took 1:34 h in general, and the
mean shrinkage rate with standard deviation were $(-3.8\pm1.0)$ nm h$^{-1}$. The shrinkage of particles was mostly linked to changes
in local atmospheric conditions, especially in global radiation and gas-phase $H_2SO_4$ proxy, or in few cases, to atmospheric
mixing. Some indirect results indicate that variations in the formation and growth rates of nucleated particles during their
atmospheric transport could be a driving force of shrinkage for particles of very small sizes and on specific occasions.

## 1 Introduction and objectives

It is increasingly recognised that atmospheric new particle formation (NPF) and consecutive particle growth events are relevant
in urban environments as well (e.g., Alam et al., 2003; Wehner et al., 2004; Borsós et al., 2012; Dall'Osto et al., 2013; Xiao et
al., 2015). The events observed in cities can be interconnected to regional NPF (Salma et al., 2016), and, therefore, their
occurrence frequency is also considerable (up to 30–35% on an annual scale), and the process often shows a remarkable
seasonal dependency with a monthly maximum of up to 60–75% in spring. New particle formation can increase the pre-
existing particle number concentrations by a factor of 2–3 (Salma et al., 2011a). As an overall effect, daily mean relative
contributions of NPF events to ultrafine (UF) particles (with diameter $d<100$ nm) for the near-city background, city centre and
street canyon in a Central European city in spring were estimated to 42%, 30% and 23%, respectively (Salma et al., 2014).
These contributions are comparable to individual high-temperature emission sources such as automotive road traffic exhaust,
common industrial combustion processes, cooking or residential heating at ordinary distances from the sources. This implicates
that NPF in urban environments can also affect the public health. Inhalation of UF particles represents an excess health risk
relative to coarse or fine particles of the same or similar chemical composition (Oberdörster et al., 2005). The specific effects
are mainly associated with large number of insoluble particles deposited in the respiratory system and their large total surface
area (Braakhuis et al., 2014; Salma et al., 2015). Nucleated particles can also impact urban climate and heat island in cities
through influencing cloud formation and properties if they grow in size to cloud condensation nuclei (Merikanto et al., 2009;
Kerminen et al., 2012).





New particle formation and growth process is superimposed on high-temperature emission sources and atmospheric transformation processes, which can – in particular in cities – complicate the identification and classification of NPF events. Larger particle number concentration levels and its fluctuation, wide variety, spatial distribution and temporal variability of

emission sources of UF particles, and the fact that the Aitken-mode particles usually have larger abundance and smaller median diameters in cities than at remote sites are the main disturbing or complicating factors. Studies on NPF processes require long and continuous atmospheric measurements of aerosol, air pollutant gases and meteorological variables. Such experimental data sets have been becoming gradually available for some cities in the world. It is increasingly desired that the peculiarities and specific features of urban NPF are systematic overviewed from several aspects. The major objectives of this paper are 1)

to present a new urban research facility, Budapest platform for Aerosol Research and Training (BpART, URL: http://salma.web.elte.hu/BpArt), 2) to discuss its surrounding air flow patterns and spatial representativity, 3) to introduce a simple correction method for particle diffusion losses in a major measuring system of UF particles, 4) to demonstrate and evaluate various types of particle formation and growth processes specifically in cities, 5) to discuss the reasons for their occurrence, and 6) to investigate and quantify the shrinkage of newly formed particles.

## 2 Methods

### 2.1 Measurement sites and time intervals

The experimental work was realised in Budapest, Hungary. Most measurements were performed at the BpART facility (Fig. 1). The platform is to serve and advance the research of atmospheric aerosols through complex surface based and satellite born measurements, as well as to promote the education and training of PhD students and young researchers. It is based on an

insulated metal container with internal dimensions of 2.00 m (width), 2.80 m (length) and 2.10 m (height) dedicated to on-line aerosol measurements and aerosol sample collections. It is located on the second floor balcony of the Northern block of the Lágymányos Campus, Eötvös University. The WGS84 coordinates (altitude, longitude and height) of the NW upper corner of the container are: N 47° 28' 29.9", E 19° 3' 44.6", 115.2 m above mean see level (a.s.l.). Its rooftop level is located in a height of 11.0 m above the street of the closest road. The distance of the container from the right bank of the river Danube is

approximately 85 m. The platform is equipped with a split air conditioning facility, an electrical heating device, a heat barrier curtain at the door, and a local area computer network. There are two independent lines for 230 V, 50 Hz 20 A AC supply available for the instruments. The facility has been in continuous operation since November 2013. Experimental data obtained for two 1-year long time interval from 13–11–2013 to 12–11–2014, and from 13–11–2014 to 12–11–2015 were considered in the present study.


Sampling inlets and sensors are set up at heights between 80 and 150 cm above the rooftop level (Fig. 1a). The platform has several waterproof vertical inlets through the ceiling and some horizontal inlets with diameters of 64, 10, 8 and 6 mm. The aerosol instruments available include a Filter Dynamics Measurement System Tapered Element Oscillating Microbalance (FDMS-TEOM 1400a, Rupprecht and Patashnick, USA), RT-OC/EC analyser (Sunset Laboratory, USA), Differential

Mobility Particle Sizer (DMPS), Stacked Filter Unit (SFU) sampler, Micro-Orifice Uniform-Deposit Impactor (MOUDI), nano-MOUDI sampler (both MSP, USA) and high-volume dichotomous sampler (HiVol virtual impactor). The air pumped inside and processed by the instruments is forwarded to the outdoors via closed internal and external tubing to a distance from the receptor site of at least 3 m in the prevailing downwind direction. Meteorological data are available from both a regular Urban Climatological Station (station number: 44505, name: Budapest Lagymanyos) of the Hungarian Meteorological Service

operated at a height of 10 m above the roof level of the building (at a height of 39 m above the ground) in a distance of about 70 m from the BpART facility, and from a simpler on-site meteorological station. Aerosol optical thickness data can be





retrieved via satellite receivers, which are located next to the regular meteorological station, from the Moderate Resolution Imaging Spectroradiometer (MODIS) on the Terra satellite.

Some additional experimental data utilised in the present paper were measured either in near-city background on the western
border of the city in a wooded area (latitude 47° 30' 1.4" N, longitude 18° 57' 46.3" E, altitude 478 m a.s.l.) from 18–01–2012 to 17–01–2013, or in a regular street canyon (Rákóczi Street, latitude 47° 29' 39.4" N, longitude 19°03' 36.3" E, altitude 111 m a.s.l) situated in the city centre from 28–03 to 31–05–2011 (Salma et al., 2014). The background is expected to represent the air masses entering the city since the prevailing wind direction is NW.

### 2.2 Experimental methods

The principal measuring system for the present study is flow-switching type DMPS. Its main components are radioactive bipolar charger, Nafion semi-permeable membrane dryer, 28-cm long Hauke-type differential mobility analyser and a butanol-based condensation particle counter (TSI CPC3775, labelled as CPC1). Particles with an electrical mobility diameter from 6 to 1000 nm are recorded in their dry state (with a typical relative humidity <30%) in 30 channels. Sample flow is 2.0 L min$^{-1}$ in high-flow mode, and 0.3 L min$^{-1}$ in low-flow mode with sheath air flows 10 times larger than the sample flows. Time
resolution of the measurement is approximately 8 min. There were a weather shield and insect net only adopted to the inlet. The DMPS measurements were performed according to the recommendations of the international technical standards (Wiedensohler et al., 2012).

Particle transport losses in the DMPS system were determined in an inter-comparison exercise by using an identical
comparative CPC (labelled as CPC2, TSI model 3775) in two steps. First, the CPC ordinary built in the DMPS system (CPC1) was compared to CPC2 to check their coherent functioning by operating them with an inlet sample flow of 0.3 L min$^{-1}$ (low-flow mode) in parallel from 26 to 27 August 2015. A data averaging time interval of 5 s was chosen because the measuring time for a single channel of the DMPS is set to 5 s. There were ca. 18 thousand data pairs obtained in this way. Operation of the CPCs are switched automatically and smoothly from live-time corrected single particle counting mode to photometric
mode at a concentration around 50000 cm$^{-3}$ (Operation and Service Manual, 2007). As the second step of the exercise, the DMPS system and CPC2 were operated in parallel from 27 August to 3 September 2015. The individual concentration data measured by CPC2 (using an averaging time of 5 s) were averaged for the actual measuring time intervals of the DMPS system (for cycle times of ca. 8 min), while the concentrations obtained in the individual channels of the DMPS were summed up. There were ca. 1160 data pairs derived in this way. Non-nucleated time intervals were only considered in the latter inter-
comparison step to minimise the effect of non-identical lower diameter measuring limit of the DMPS (6 nm) and CPC2 (4 nm). The sampling lines for the CPCs and DMPS were prepared in an identical way, and were set up at the BpART in a distance of 1 m from each other.

Standardised meteorological measurements of air temperature ($T$), relative humidity (RH), wind speed (WS), wind direction
(WD) and precipitation (Prec) were recorded with a time resolution of 10 min. Global radiation (GRad) was calculated from the measured meteorological data by methodology of Holtslag and Van Ulden (Weidinger et al., 2008). Wind roses were based on the modern Beaufort scale of 0 (WS<0.3 m s$^{-1}$), 1 (0.3–1.5 m s$^{-1}$), 2 (1.6–3.3 m s$^{-1}$), 3 (3.4–5.5 m s$^{-1}$), 4 (5.6–7.9 m s$^{-1}$), 5 (8.0–10.7 m s$^{-1}$), 6 (10.8–13.8 m s$^{-1}$) and 7 (13.9–17.1 m s$^{-1}$). Concentration of pollutant gases were obtained from the closest measurement station of the National Air Quality Network in Budapest (BP6) in a distance of 1.6 km from the urban site in the
upwind prevailing direction. Concentrations of SO$_2$ and O$_3$ – considered here – were determined by UV fluorescence (Ysselbach 43C) and UV absorption (Ysselbach 49C, both Austria), respectively with a time resolution of 1 h.



### 2.3 Data evaluation and modelling

Evaluation of the measured DMPS data was performed according to the procedure protocol recommended by Kulmala et al. (2012). Mathematically inverted size distributions were utilised for calculating particle number concentrations in the particle diameter ranges from 6 to 25 nm ($N_{6-25}$), from 6 to 100 nm ($N_{6-100}$), from 100 to 1000 nm ($N_{100-1000}$) and from 6 to 1000 nm

($N$). The individual size distributions were fitted by lognormal functions using the DoFit algorithm (Hussein et al., 2004) to determine modal parameters including number median diameters (NMDs). The individual concentration data were also utilised to generate particle number size distribution surface plots showing jointly the variation in particle diameter and particle number concentration density in time. Identification and classification of NPF processes was accomplished by using an algorithm of Dal Maso et al. (2005) on a day-to-day basis into the following main classes: NPF event days, non-event days, days with

undefined character, and days with missing data for more than 4 h in the late morning. Frequency of events (for a month or a year) was determined as the ratio of the number of event days to the total number of relevant days. A subset of NPF events with uninterrupted evolution, which are called quantifiable (class-1A) events, were further distinguished to calculate their dynamic and timing properties. Growth rate (GR) of particles for a size interval of 6–25 nm was calculated by log-normal distribution function method (Kulmala et al., 2012). Condensation sink (CS) for $H_2SO_4$ molecules onto the surface of existing

dry aerosol particles was computed according to Dal Maso et al. (2005). Shrinkage rate (SR) of the nucleation-mode particles was derived as the slope of the fitted linear line of several NMD data in its time evolution plot, similarly to GRs. Gas-phase $H_2SO_4$ proxy values were calculated for GRad>10 W m$^{-2}$ according to Petäjä et al. (2009). The earliest estimated time of the beginning of a nucleation (time parameter $t_1$), the latest estimated time of the beginning of a nucleation (time parameter $t_2$), and the ending time of the particle growth process (time parameter $t_e$) were determined for each quantifiable NPF event

(Németh and Salma, 2014). The time parameter $t_1$ also includes the time shift that accounts for the particle growth from the stable neutral cluster mode at (1.5±0.4) nm (Kulmala et al., 2013) to the smallest detectable diameter limit of the DMPS systems (6 nm). Current local time (CLT=UTC+1 or daylight saving time of UTC+2) was chosen as the time scale for the evaluations.

Air flow and dispersion of aerosol particles in the surroundings of the BpART research facility were investigated by using a three-dimensional Computational Fluid Dynamics (CFD) model of the adjacent territory in order to estimate the representativity of results and conclusions obtained at a single measurement point. The model includes a detailed geometrical description of the host university building (where the platform is located) as well as a simplified representations of the neighbouring buildings and vegetation within a 600-m radius area (Fig. 1c). For specification of the boundary conditions, a

300-m wide relaxation zone was added to the periphery of the orographic model artificially, which equalizes the terrain elevation height. The vertical profiles of velocity and turbulent quantities were specified at cylindrical shape lateral boundary. The domain height was set at 200 m, and symmetrical boundary condition was used at the top of the domain. The numerical solution was developed in ANSYS-FLUENT solver (version 16.2) by using realizable $k$-$\varepsilon$ turbulence model. Steady isothermal flow was assumed from 8 different wind sectors of 45° angular raster. Vertical distribution of the inflow velocity and turbulence

quantities were determined from a supplementary 2D boundary layer model on the way to obtain the measured median WS in the 8 wind sectors above the rooftop level. Both summer (vegetation with leaves) and winter (no vegetation) canopy conditions were taken into account. Porous resistance of the summer canopy for the vegetation was calculated from the leaf area index of the dominant flora (5.3 m$^2$/m$^2$; Asner et al., 2003), while hydraulic resistance of the winter canopy was neglected. The observation volume element for the CFD modelling was set by the roof area of the BpART facility and by a height of 1 m

above its rooftop level.





## 3 Results

Ranges and medians of the daily median particle number concentrations and UF particle contribution for central Budapest are summarised in Table 1 for general orientation.

### 3.1 Correction of transport losses

Evaluation of the concentration data measured by CPC1 (ordinary built into the DMPS system) and CPC2 (comparative instrument) is shown in Fig. 2. The agreement between the two CPCs near and above the limiting range between the single particle counting mode and photometric mode is not acceptable. Both the scatter plot and concentration ratios show tendentious deviations. This is most likely caused by differences in the live-time correction at high counting rates, and in the actual transition ranges between the operating modes for the two instruments, which are based on manufacture's calibration.

Fortunately, such high concentrations do not occur when the CPC measures size-separated particle fractions in the channels of the DMPS system (see later). Considering the data below 30000 $cm^{-3}$, the slope of the regression line for the two CPCs was 1.034, and the mean concentration ratio with its standard deviation are 1.11±0.12. The latter quantification seems to be in accordance with the nominal specification of CPCs, which means an agreement better than 10% between the instruments with a data averaging interval of >30 s. There was no systematic deviation in the ratios with concentration under 30000 $cm^{-1}$. The

individual ratios, however, showed better agreement for night-time (slowly changing) values than for daytime (more dynamic) concentrations. For the time interval from 23:00 to 4:00, the coefficient of determination increased to $R^2$=0.997, the mean concentration ratio and its standard deviation decreased to 1.03±0.02, and the slope and intercept of the regression line were 1.006 and 274 $cm^{-1}$, respectively. In order to quantify the time response of the CPCs, an identical high-efficiency particulate-free air filter (HEPA filter capsule, Pall, USA) was adjusted to them step wisely. Temporal variation of the measured

concentrations in this time interval is shown in Fig. 3. It is seen that CPC1 reacted on the concentration decrease faster than CPC2 (although both satisfied the certified response time of 5 s to 95% of concentration step change). For CPC2, there was a fall of 4 orders of magnitude which was followed by a slower decrease. The latter change could be expressed by a relaxation time of 1.6 min. It is concluded that the difference of the timing properties of the CPCs can contribute to the moderate differences in the measured concentrations between the 2 CPCs.

Evaluation of the data obtained from the DMPS system and CPC2 is shown in Fig. 4. It is worth noting that no individually measured DMPS concentration data (in a channel) was larger than 10000 $cm^{-3}$ despite the fact that the comparison exercise was performed mostly under anti-cyclonic weather conditions, which are usually associated with more polluted air. It can be seen in Fig. 4 that the DMPS measures systematically smaller total concentrations than the CPC. This is due to larger transport

losses of particles within the longer tubing and other parts of the DMPS system. Mean CPC/DMPS concentration ratio and its standard deviation were 1.26±0.18, and the median ratio was 1.22. The ratios did not indicate any tendency and resembled fluctuations both in time and with concentration. As a result of this inter-comparison exercise, the median CPC/DMPS concentration ratio is adopted as correction factor for the total particle number concentrations obtained by the DMPS system. It is worth noting that the intercept of the regression line and its standard deviation were (−163±149) $cm^{-3}$, which indicate that

the smallest reliably measurable total concentration with the DMPS is approximately 500 $cm^{-3}$. The smallest total concentrations obtained in 3 years of operation in central Budapest were around 1000 $cm^{-3}$, which confirms that the actual DMPS system is capable and appropriate instrument under the atmospheric conditions encountered.

### 3.2 Meteorological conditions and air flow structure around BpART

Averages of some basic meteorological data inside and outdoors the BpART facility (based on ca. 50 thousand measured data

pairs each) are summarize in Table 2 for the 1-year long time interval in 2013–2014. They demonstrate that the conditions inside the BpART can be well maintained (for $T$ and RH within ±5 °C and ca. ±20%, respectively) in ordinary operation for





an annual run, and that they enable high-quality experimental work at the platform. The conditions inside can be kept within even a narrower range for shorter intervals. The outdoor mean $T$ is somewhat larger (by 1.3 °C), while the mean WS is somewhat smaller (by 0.7 m s$^{-1}$) than for a climatological site in the city, which are most likely caused by local urban impacts, such as the heating and shadowing effects of the host and neighbouring buildings, and by the close vicinity of a large water

reservoir (Danube). Wind roses derived for data sets obtained both above rooftop level and on-site the BpART facility are shown in Fig. 5 for 2013–2014. The data for 2014–2015 yielded similar results. It can be seen that the wind roses are rather different. The wind rose for the rooftop level shows major WDs in N and NE due to the modifying effects of the natural orogeny (mainly Buda Mountains) and of the wind channel, which is often formed above the river Danube, on the prevailing synoptic wind (which is N-NW). At the same time, the wind rose for the on-site data shows 3 other major directions, namely

W, SE and S. This latter property is largely governed by the effects of the actual urban canopy on the local urban air flow. The wind rose for the rooftop level assuming that the major on-site WD of W and WS>0.3 m s$^{-1}$ is also displayed in Fig. 5. It indicates that the arrival direction on a larger horizontal scale was obviously and almost exclusively N. Conditional wind roses for the other two major on-site directions of SE and E showed almost identical directions at the rooftop level, with a somewhat more emphasis on the direction SE. The former discrepancy between and the latter similarities in the wind roses at the 2

locations can be mainly explained by the effect of the actual urban construction. The interpretations based on the wind roses are further supported by air flow field modelling results (see below). Taking into account these relationships, the on-site meteorological station can be very well utilised for prompt orientation in ordinary or extreme meteorological situations, and for making immediate decisions in experiments at the platform.

Streamlines (trajectories of massless particles) were traced back by using a reversed flow field from the observation volume element to a distance of 600 m in radius. Histogram of the starting point height for the 8 major wind sectors are displayed in Fig. 6. It reveals that the effective sampling height changes with WD. The most probable height of origin was 3–8 m for most WDs (of N, NE, E, SE and NW). In the case of S, SW and W wind directions, however, when the observation volume element falls onto the lee side of the host building, most air parcels arrived from a height of 25–40 m. This suggests that the air flow

around the host building has a major impact on the sampling height. No substantial dependency on the canopy drag was found. Flow structure around the BpART facility for the 8 major wind sectors in summer are shown in Fig. 7. The path lines indicate that the host and neighbouring buildings together with the natural orography play an important role in the near field dispersion process. Velocity distributions revealed that the vegetation causes a minor upward displacement of the streamlines, as expected, and that the near-ground velocity decreases. The air flows arriving from the directions N, NE, E and SE bring in air

masses directly from further distances to the BpART facility. When the wind blows from S, SW, W or NW, the effects of eddies and turbulence at the platform cannot be neglected, and the on-site WSs are usually decreased and on-site WDs usually substantially modified. The details and features of the air flow structure should be taken into consideration when interpreting horizontal extension of atmospheric processes and phenomena.

### 3.3 Growth types of nucleated particle in cities

Many NPF and consecutive particle growth processes were observed as banana-shaped plots (Fig. 8a), which indicate that the measurement site was often exposed to spatially homogenous air masses (Kivekäs et al., 2016). This shape was usually associated with regional nucleation phenomenon in the Carpathian Basin (Salma et al., 2016), and occurred under relatively clean atmospheric conditions. Some plots appeared with a narrow onset as NPF burst. A few similar banana plots were also observed in a rather polluted air. The basic preconditions of NPF events are realised by competing source and sink terms, and,

therefore, NPF can occur even in polluted environments (with large condensation and scavenging sinks) if the sources for condensable chemical species are even larger, and the other conditions are also favourable (Fig. 8b). Changes in the intensity of local emission sources can cause considerable fluctuations of the nucleation-mode NMD in the overlapping diameter range.





A flow-mediated banana plot (Fig. 8c) was observed in the street canyon, which has one end at the bank of the river Danube, and the other end in the city. There are two essential horizontal air flows possible inside the canyon: either from or toward the river, which lead to rather diverse atmospheric conditions. In some cases, NPF events were clearly recognisable as banana plots for several hours when the wind blew from the river (bringing in cleaner and regionally more representative air masses),

and the curve disappeared when the air flow changed to the opposite direction (introducing more polluted air masses, in which, there was no NPF and growth process going on, or which inhibited the ongoing process).

Figure 9 shows NPF and particle growth events with multiple consecutive onsets. During the relevant time intervals, the properties which are related to NPF events varied substantially and rapidly in time, while the concentration $N_{100-1000}$, which

represents a larger spatial region, and the local WD, WS and GRad data, the path of the air mass stayed largely constant (Salma et al., 2016). The first onset in Fig. 9a was associated to regional NPF, and the later onsets occurred in a more polluted air, which was unambiguously of urban or local origin. This takes place, for instance, by mixing regional and urban air parcels that exhibit different properties, and which is mainly governed by local planetary boundary layer (PBL) dynamics and urban heat island effects (Kulmala et al., 2005; Hirsikko et al., 2013). A NPF with up to four onsets are shown in Fig. 9b although

no of them was developed as a banana plot. Multiple onsets, which occur close enough to each other in time, can result in overlaps, and can be a reasons for banana plots with a broad onset extending up to 4–6 h (Fig. 9c). Determining dynamic properties for such a shape could be challenging and associated with larger uncertainty.

Figure 10a shows a NPF with an uninterrupted particle growth which was limited in time. The duration of the growth phase

can be variable. In some cases, the particle growth was followed by a reverse process, i.e., by continuous decrease in nucleation-mode NMD as displayed in Fig. 10b. This combination results in arch-shape surface plot. In some cases, the nucleated particles shrank back to the smallest measurable diameter limit for the DMPS of 6 nm. The actual plot (Fig. 10b) was produced in a blizzard when $SO_2$, WD and trajectory path stayed mostly constant, daily median WS was 6 m s$^{-1}$, and $T$ and RH decreased from a daily median of 3.0 °C on the previous day to –2.2 °C on the day of the NPF, and from 99% to 64%,

respectively. It is mentioned that NPF occurred on the next day as well, which further supports the correct identification of the event. Arc-shape surface plots contain information about the nucleated particles, and, therefore, they are further evaluated and discussed in Sect. 3.4.

Size distribution surface plot generated by NPF and growth events are usually superimposed on the effects of other urban

emission sources and transformation processes. Some of them, such as vehicular road traffic, happen almost regularly since they often follow diurnal activity pattern of inhabitants on workdays (Morawska et al., 2008; Salma et al., 2011b). Some other ordinary substantial emission sources, such as boat traffic on rivers in cities or diesel-driven single heavy-duty tracks or buses, cause sudden and considerable changes (red stripes) on surface plots in an irregular or occasional manner. Furthermore, number concentrations can be also modified very much or sometimes even in a determinative way by local meteorology and

atmospheric mixing. A typical surface plot reflecting some of these effects is shown in Fig. 11a. Impacts of residential heating and combustion activities mainly in winter evenings can also add to the situation. Figure 11b shows a surface plot for a very specific activity, namely an extensive grass cutting in the neighbouring park of the BpART facility. It was likely generated by organic emissions from the lysed vegetation tissue combined with the exhaust emissions from the related internal combustion engines, and it is expected to affect a very local area only, and should not be classified as NPF on a larger spatial scale. This

latter example points to the necessity of personal observations and their regular logging.



### 3.4 Shrinkage of nucleated particles

There were 178 quantifiable NPF and consecutive particle growth events identified in Budapest in 4 years. Of them, 4.5% (8 days) yielded well-developed uninterrupted arch-shape surface plots. Most of them occurred in spring (50%) and summer (25%), although the number of cases was rather limited to arrive at solid conclusions. There were arc-shape surface plots

observed in the other NPF classes than quantifiable events as well but their rigorous evaluation was hindered by their non-unambiguous or interrupted growth or shrinkage phases, and fluctuating data. Decreasing process of the Aitken-mode NMD was also detected on some non-nucleation days. The former case was only evaluated because it supplies representative and reliable information on nucleated particles. The SR varied from –4.8 to –2.3 nm h$^{-1}$ with a mean and standard deviation of (–3.8±1.0) nm h$^{-1}$. The values are comparable to the data observed earlier (Yao et al., 2010; Young et al., 2013; Skrabalova et

al., 2015). They are close, in particular to the rates that represent shrinkage due to changing meteorological conditions (Cusack et al., 2013; Skrabalova et al., 2015), which is in line with our further evaluation (see later). The shrinkage driven by changes in the atmospheric mixing are usually more rapid (Yao et al., 2010). It is noted that the GR for the related NPF events changed from 3.4 to 6.1 nm h$^{-1}$ with a mean and standard deviation of (4.5±1.2) nm h$^{-1}$. It was found earlier that particle GRs show seasonal dependency in the Carpathian Basin (Yli-Juuti et al., 2009; Salma et al., 2011a). A similar tendency for SRs could

not be established so far due to the small number of arc-shape surface plots.

The shrinkage of newly formed particles was earlier attributed to evaporation of reversible condensed chemical species (for urban environments, Yao et al., 2010; Backman et al., 2012; Young et al., 2013; Skrabalova et al., 2015; for a regional background, Cusack et al., 2013). Its primary causes were explained either by changes in meteorological conditions or in

atmospheric vertical mixing. They both lead to decreased atmospheric concentration of vapours below their super-saturation. Decrease of the nucleation-mode NMD due to coagulation and collapse of structured particles with void volume were not considered, although the latter possibility does not seem very realistic for nucleated particles. Recent studies reported that major chemical compounds of particles grown after NPF are organics (including carboxylic and hydroxyl acids, N-containing organic compounds; Smith et al., 2008), nitrate, sulphate and ammonium ions (Bzdek et al., 2012). Some of them can be indeed

semi-volatile. Another study (Kivekäs et al., 2016) concluded, however, that this explanation needs further elaboration since newly formed particles in their first stage are expected to consist of compounds having low or extremely low vapour pressure (Ehn et al., 2014). Moreover, freshly nucleated particles with a median mobility diameter of ca. 20 nm collected on transmission electron microscopy (TEM) grids survived high-vacuum conditions in the TEM column, although for limited time (Németh et al., 2015). They only evaporated when exposed to the electron beam. In addition, a simplified atmospheric

transport model which considered temporal, spatial or both variations in formation rate ($J$) and GR during the transport, could yield various shapes of surface pot for a fixed site including those showing decrease in NMD and $N$ even if the local conditions did not vary.

In order to investigate the effect of the local meteorology and some air pollutants, a growth phase and a shrinkage phase

(corresponding to the increase or decrease in the nucleation-mode NMD, respectively) of the arc-shape plots were selected by linear approximation. Median values and change rates of the relevant aerosol-related and meteorological properties over these time intervals were derived separately. Their mean values for the identified 8 days together with the mean growth/shrinkage ratios are summarised in Table 3. It is seen that evident particle growth and shrinkage happened for similar time intervals of ca. 1:30–2:00 h. Concentrations $N_{6-25}$, $N_{6-100}$ and $N$ increased substantially (by 45–70%) during the growth phase as expected,

and they decreased in general by 30–50% during the shrinkage phase. As far as the individual cases are concerned, temporal variation of $N_{6-25}$ during the shrinkage phase was decreasing in almost all cases. In addition, median $N_{6-25}$ concentration over the shrinkage phase was usually smaller than that over the growth phase. These can be explained by increased coagulation sink for the shrinking particles. On few (namely 2) days, an opposite tendency, thus increasing $N_{6-25}$ over the shrinkage phase was



observed. This could be explained by contribution of urban emission sources into the upper end of the specified diameter interval (around 25 nm), which could not be rejected on these specific days, or to the changes in local meteorology, which could modify atmospheric concentrations. Variations in $N_{100-1000}$, representing a larger horizontal scale and background aerosol over the growth and shrinkage phases were negligible. Concentration of $SO_2$ precursor gas was also rather constant during

both the growth and shrinkage phases, which suggests that $SO_2$ does not represent serious limitations to the shrinkage process. Modest increase in CS during the growth phase can be explained by increased surface area of the growing particles and particle number concentrations. In the shrinkage phase, CS was decreasing more rapidly than for the growth. The mean $T$ for the time interval of interest was between 16 and 18 °C for the present arc-shape plots, but for several days, it was well below 10 °C. The plot presented in Fig. 10b even occurred at daily median $T$ as low as −2.0 °C. This is an important extension to a previous

conclusion (Skrabalova et al., 2015) according to which under ca. 12 °C the evaporation of atmospherically relevant organics and $NH_4NO_3$ is significantly hindered. In the present case, other mechanisms, for instance sudden changes in atmospheric mixing, or more importantly, variations in $J$ and GR values during the atmospheric transport should be considered. It is noted in addition that there were no substantial changes in $T$ and RH between the growth and shrinkage phases, which imply that thermal evaporation or fragmentation of, or water evaporation from particles cannot be considered as realistic explanations.

Changes of $O_3$, GRad, Proxy, RH, $T$ and partially WS are biased by their ordinary diurnal cycling or pattern, while the atmospheric concentrations are also influenced by PBL dynamics and mixing intensity (Hamed et al., 2010). Global radiation and gas-phase $H_2SO_4$ proxy showed the largest relative change when taking into account by both the mean growth/shrinkage ratios and substantial change rates during the shrinkage phase. For them, there can be further relationships – in addition to the bias – between the variables and the physicochemical processes undergoing within the particles during the shrinkage. This can

include decreased photochemical activity, or decreased concentration of condensable chemical species leading eventually to exhausted atmospheric super-saturation. The shrinkage could not be, however, fully explained by local atmospheric parameters. Relatively large time resolution (1 h) of GRad in the present work, unfortunately, did not allow to investigate the evaporation dynamics.

## 4 Conclusions

Chemical and physics situations in urban atmospheric environments can be rather complex and variable both in space and time. They are also affected with close surroundings, constructions and orogeny. Long-term, continuous and representative measurements, which are basis of contemporary aerosol studies, require dedicated research infrastructure and devices. This also means that supplementary and inter-disciplinary studies need to be performed at these measurement sites. Here we demonstrated a joint detailed study in atmospheric chemistry/physics, meteorology and fluid dynamics for this purpose. The

results and conclusion achieved represent firm background for later interpretations and conclusions. Various shapes of particle growth process observed in 4-year long measurements were presented and their classifications were discussed in detail. Misleading conclusions could be drawn without considering the actual activities around the measurement sites in cities, in particular in combination with transformation and other disturbing effects, and fluctuating data. The particle diameter range <10 nm of the measuring (DMPS/SMPS) systems seems invaluable for correct and reliable NPF identification in urban

environments. Arch-shape surface plots provide information on freshly nucleated particles. Their shrinkage could be mostly associated to changes in local meteorology or atmospheric conditions, in particular decreased GRad and gas-phase $H_2SO_4$ proxy, and partially, by air mixing. It is also noted that the role of different chemical species to the particle growth depends on the particle size as well. Non-volatile species are mainly responsible for the formation and initial growth, while contribution of more volatile compounds increases with particle size (Zhang et al., 2004; Smith et al., 2008). This implies that both major

ideas on the reasons for particle shrinkage (i.e., evaporation or variations in $J$ and GR during the atmospheric transport) could be effective. Further insights into the shrinkage can be obtained in dedicated experiments on the evaporation dynamics and also including aerosol mass spectrometry on one side, and extended chemical transport modelling on the other side.



**Acknowledgements**

Financial support by the Hungarian Scientific Research Fund (contract K116788) is appreciated. We are grateful to Regina Hitzenberger and Anna Wonaschütz, both of the University of Vienna for providing the CPC2 for the inter-comparison exercise, and would like to thank Veronika Varga (Eötvös University) for her help in the data treatment of the inter-comparison exercise, and Mihály Vince (Budapest University of Technology and Economics) for his assistance in preparing the CFD modelling.

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





**Table 1:** Ranges and medians of daily total particle number concentrations ($N$), and mean contributions of ultrafine (UF) particles to $N$ with standard deviations in central Budapest for 1-year long intervals in 2008–2009, 2013–2014 and 2014–2015.

| Time interval | Range [$\times 10^3$ cm$^{-3}$] | Median $N$ [$\times 10^3$ cm$^{-3}$] | Mean UF/$N$ ±std. dev. [%] |
|---|---|---|---|
| 2008–2009 | 3.8–29 | 11.8 | 79±6 |
| 2013–2014 | 2.8–23 | 9.7 | 75±7 |
| 2014–2015 | 1.91–22 | 9.3 | 75±8 |



**Table 2:** Ranges, medians and means with standard deviations for air temperature ($T$), relative humidity (RH), wind speed (WS) and daily precipitation (Prec) inside and outdoors the BpART facility for 1-year long ordinary operation in 2013–2014.

| Meteorological variable/ Statistics | Inside | | Outdoors | | | |
|---|---|---|---|---|---|---|
| | $T$ [°C] | RH [%] | $T$ [°C] | RH [%] | WS [m s$^{-1}$] | Prec [mm] |
| 1%-percentile | 14.5 | 21 | −2.6 | 37 | <0.1 | 0 |
| Median | 18.2 | 39 | 14.8 | 79 | 1.4 | 0 |
| 99%-percentile | 23.1 | 66 | 33 | 99 | 6.1 | 30 |
| Mean | 18.4 | 41 | 14.3 | 77 | 1.5 | 1.6 |
| St. deviation | 1.7 | 10 | 8.4 | 17 | 1.3 | 5.0 |





**Table 3:** Means and their mean relative change rates (in an hour expressed as % of the mean) separately for the growth (G) and shrinkage (S) phases of arch-shape NPF processes, together with mean G/S ratios of medians for the duration time, nucleation-mode particle number median diameter ($NMD_{nucl}$), particle number concentrations in various size fractions ($N_{6-25}$, $N_{6-100}$, $N_{100-1000}$, $N$), condensation sink (CS), $SO_2$ concentration, $O_3$ concentration, gas-phase $H_2SO_4$ proxy value, global
5    radiation (GRad), relative humidity (RH), air temperature ($T$) and wind speed (WS).

| Variable | Growth phase | | Shrinkage phase | | Mean |
| --- | --- | --- | --- | --- | --- |
| | Mean | Change | Mean | Change | G/S ratio |
| Duration [hh:mm] | 1:52 | – | 1:33 | – | 1.2 |
| $NMD_{nucl}$ [nm] | 9.6 | 3 | 10.8 | –4 | 0.8 |
| $N_{6-25}$ [$\times10^3$ cm$^{-3}$] | 7.6 | 68 | 6.5 | –47 | 1.6 |
| $N_{6-100}$ [$\times10^3$ cm$^{-3}$] | 10.4 | 52 | 8.4 | –35 | 1.5 |
| $N_{100-1000}$ [$\times10^3$ cm$^{-3}$] | 1.5 | n.r. | 1.2 | –4 | 1.5 |
| $N$ [$\times10^3$ cm$^{-3}$] | 11.6 | 46 | 9.4 | –31 | 1.4 |
| CS [$\times10^3$ s$^{-1}$] | 0.008 | 2 | 0.006 | –8 | 1.4 |
| $SO_2$ [μg cm$^{-3}$] | 7.4 | –8 | 5.9 | 6 | – |
| $O_3$ [μg cm$^{-3}$] | 57 | 22 | 75 | 3 | 0.6 |
| Proxy [μg m$^{-5}$ W s] | 412 | 91 | 732 | –18 | 0.7 |
| GRad [W m$^{-2}$] | 143 | 95 | 463 | –14 | 0.4 |
| RH [%] | 64 | –6 | 56 | 1.1 | 1.1 |
| $T$ [°C] | 16 | 23 | 18 | –6 | 0.4 |
| WS [m s$^{-1}$] | 1.4 | –4 | 1.3 | –3 | 1.1 |

n.r. not relevant (below 1% in absolute value)





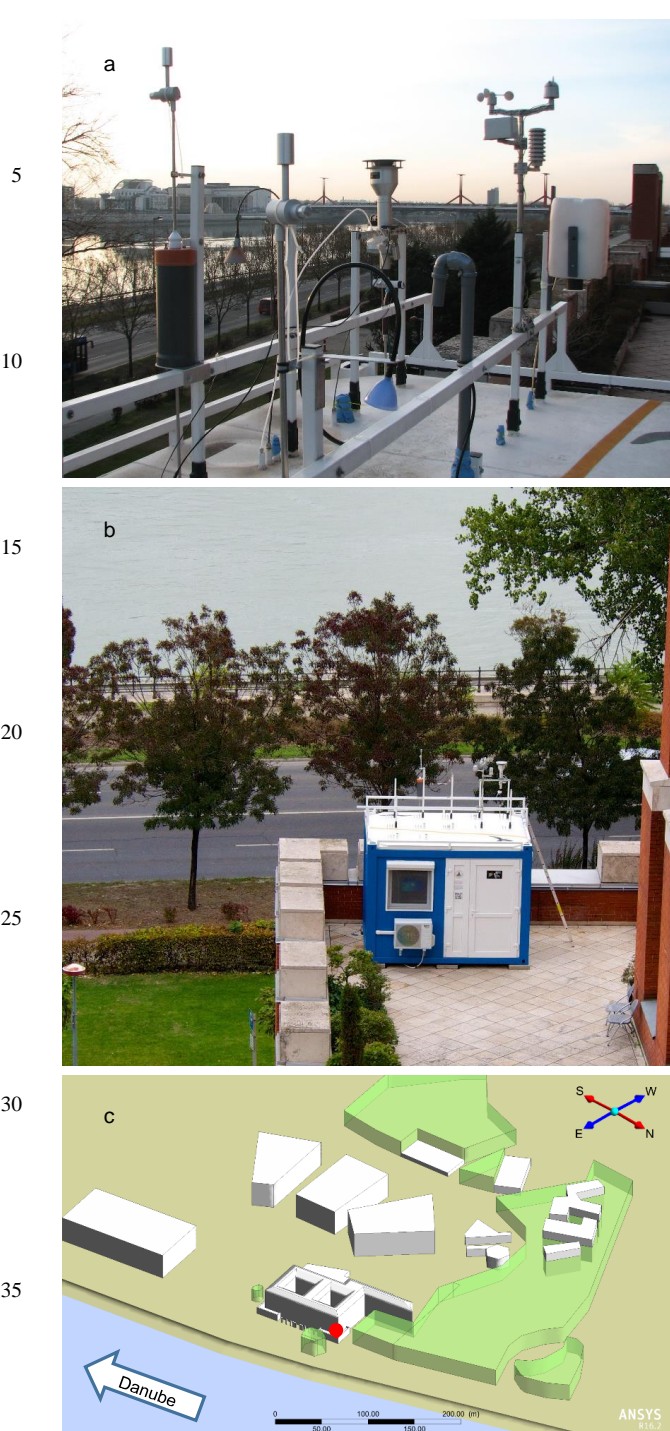

**Figure 1:** The superstructure (a) and close surroundings (b) of the BpART facility together with geometrical representation of the buildings and vegetation in a perspective view (c) utilized in the CFD modelling. The location of the platform indicated by a red dot and a compass rose are also shown in the lowest panel.





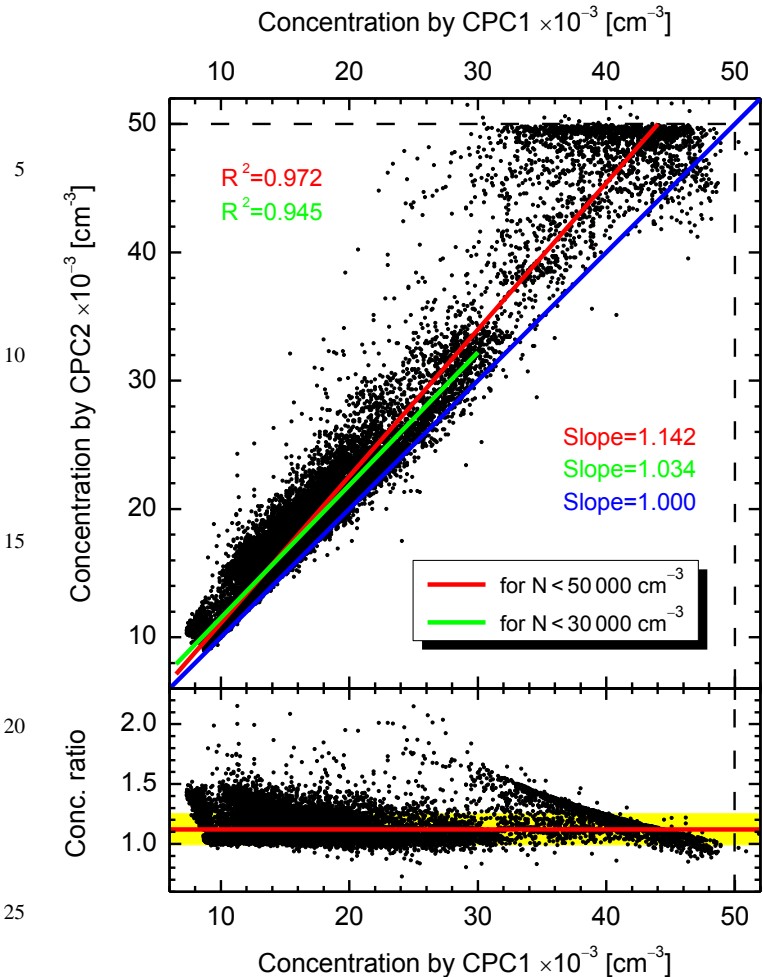

**Figure 2:** Scatter plot and CPC2/CPC1 concentration ratios for the data obtained by two identical CPCs. In the upper panel, the coefficients of determination of the data pairs ($R^2$), the regression lines and their slopes are shown for concentrations $N<50000$ and $<30000$ cm$^{-3}$ in the corresponding colours, together with the equality line. In the lower panel, the horizontal red line and yellow band indicate the mean concentration ratio and its standard deviation, respectively.





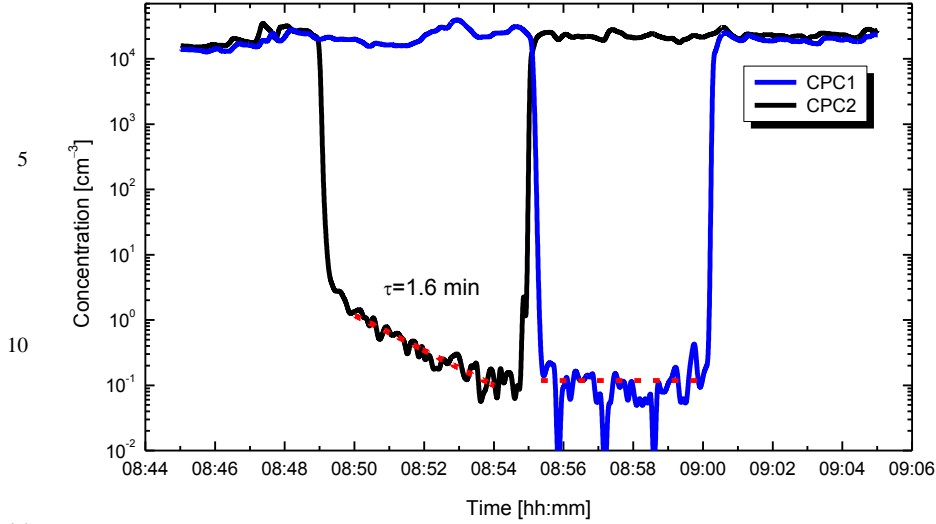

**Figure 3:** Temporal response of two identical CPCs to the concentration change caused by adjusting the same HEPA filter to them step wisely. The dashed lines show the fitted linear curves. Relaxation time τ of the slower CPC is also indicated.





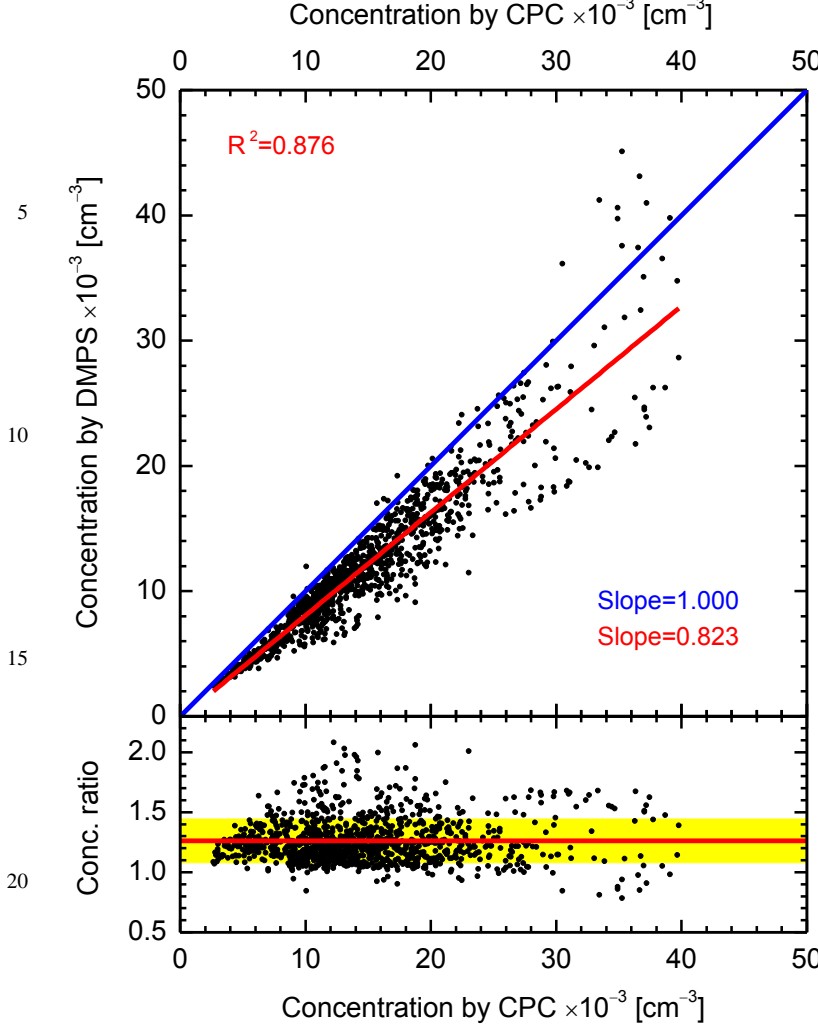

**Figure 4:** Scatter plot and CPC/DMPS concentration ratios for total particle number concentration data derived from the DMPS system and from an identical CPC. In the upper panel, the coefficient of determination of the data pairs ($R^2$), the regression line and its slope are shown together with the equality line. In the lower panel, the horizontal red line and yellow band indicate the mean concentration ratio and its standard deviation, respectively.





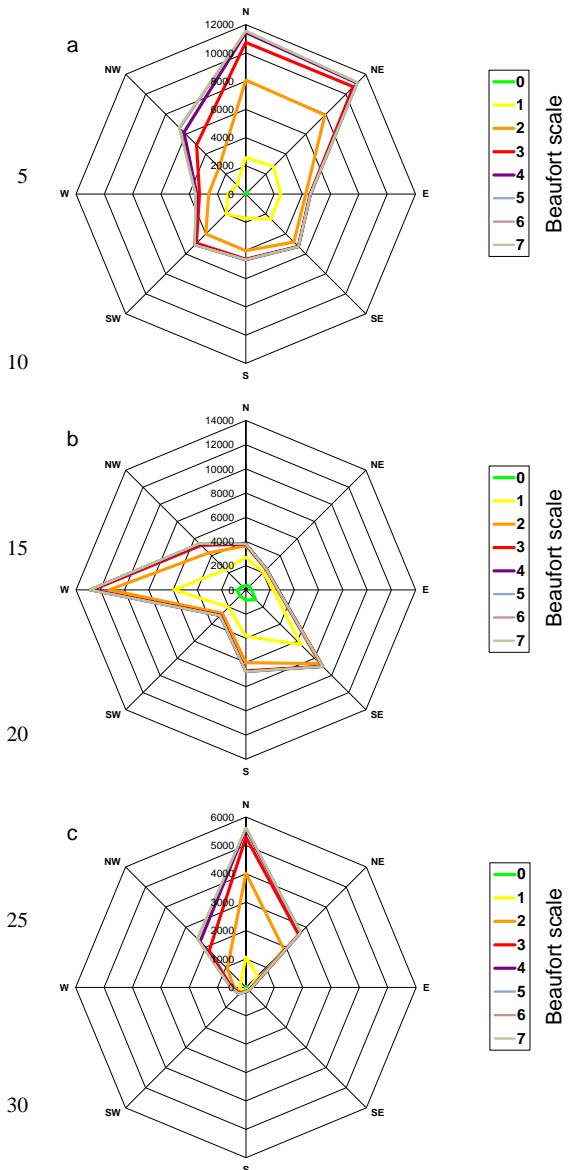

**Figure 5:** Wind roses based on the modern Beaufort scale for the data sets obtained above the rooftop level of the building (a), on-site the BpART facility (b), and for the data above the rooftop level on the condition that the WD on-site was W and its speed >0.3 m s$^{-1}$ (c). Figures on the vertical axis indicate the scale for the number of data belonging to each case.



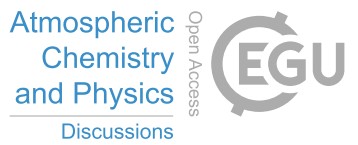

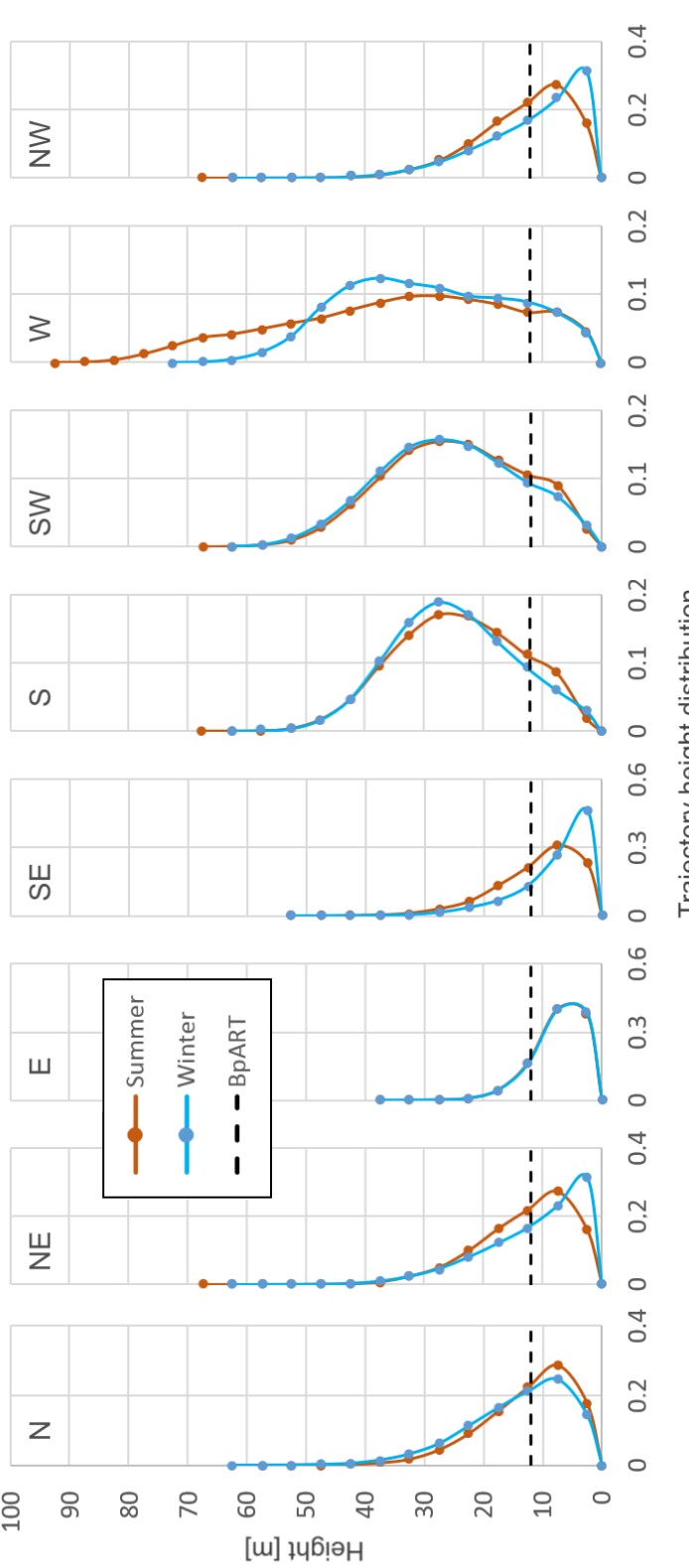

**Figure 6:** Histogram of the starting point height at a distance of 600 m from the BpART facility for 8 major wind sectors (N, NE, E, SE, S, SW, W and NW) for summer (vegetation with leaves) and winter (no vegetation). The sampling height of the platform is indicated by dashed lines.



**Figure 7:** Path lines and speeds around the BpART facility for 8 major wind sectors (N, NE, E, SE, S, SW, W and NW) in summer (vegetation with leaves) in a perspective view. The wind directions are marked by an arrow and letters, and the location of the platform is indicated by red dot in the panels.





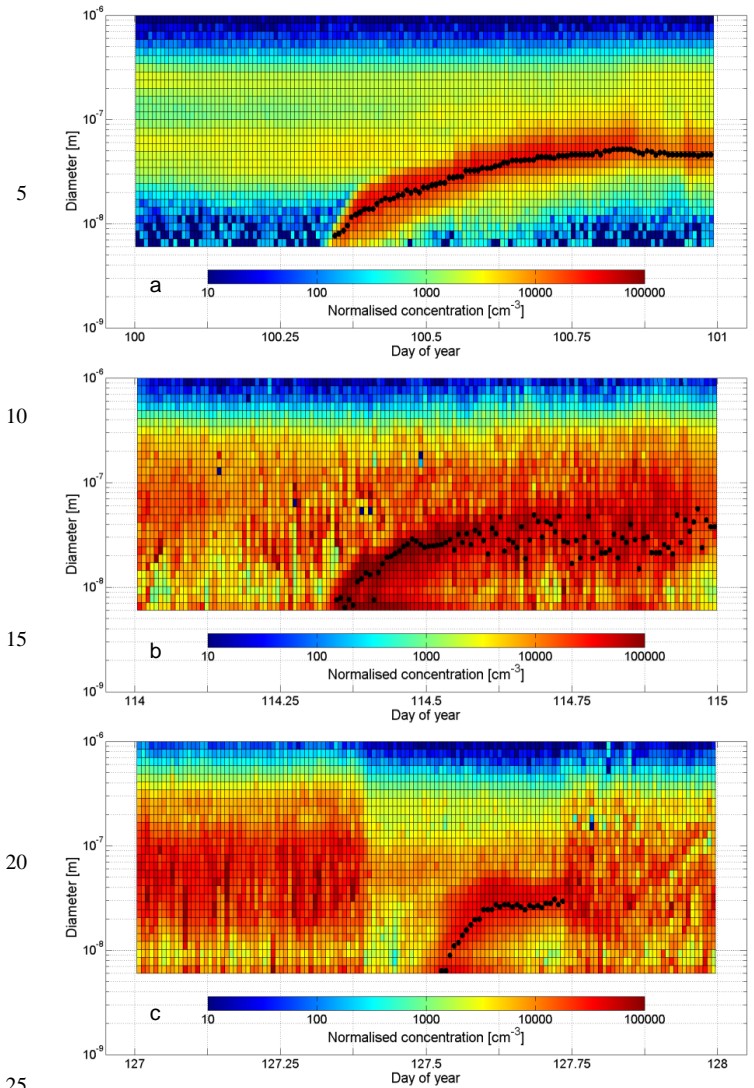

**Figure 8:** Banana-shape size distribution surface plots for regional NPF and consecutive particle growth with narrow onset (a) in the near-city background on Monday, 09–04–2012; in a polluted urban air within the street canyon (b) on Sunday, 24–04–2011; and appearing in a time window (c) in the street canyon on Saturday, 07–05–2011. Time series of the fitted particle number median mobility diameter for the nucleation mode are indicated by black dots on each panel.





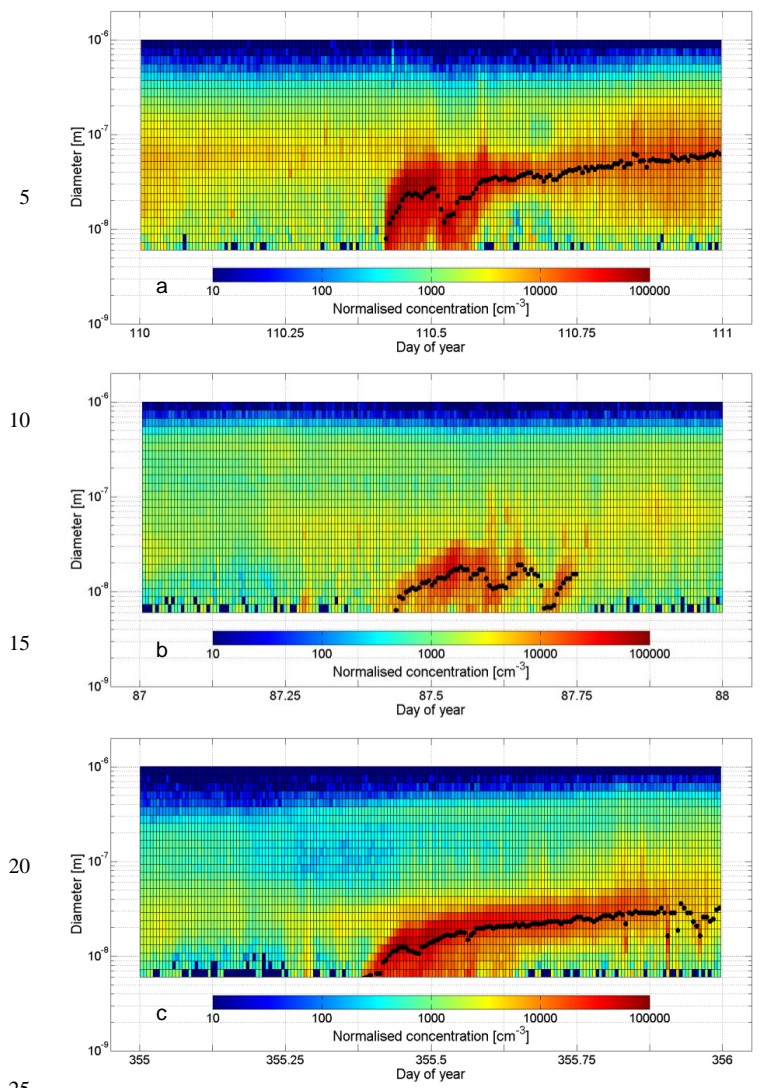

**Figure 9:** Size distribution surface plots for NPF and consecutive particle growth as banana-shape plot with double onset (a) in the city centre on Sunday, 20–04–2014; with multiple onsets (b) in the city centre on Saturday, 28–03–2015; and with broad continuous onset (c) in the city centre on Sunday, 21–12–2014. Time series of the fitted particle number median mobility diameter for the nucleation mode are indicated by black dots on each panel.

30





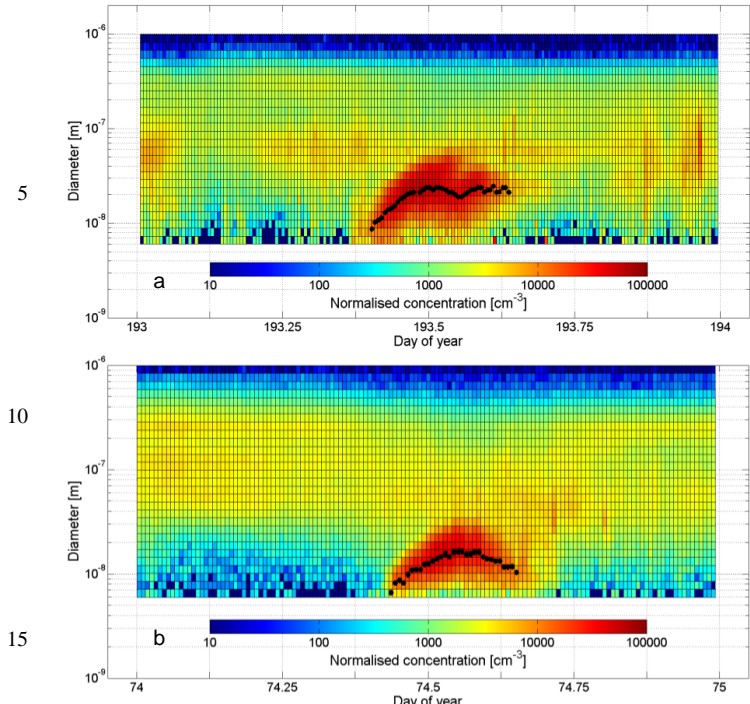

**Figure 10:** Size distribution surface plots for NPF and consecutive particle growth as banana-shape plot limited in time (a) in the city centre
on Tuesday, 12–07–2012; as arch-shape plot (b) in the city centre on Friday, 15–03–2013. Time series of the fitted particle number median
mobility diameter for the nucleation mode are indicated by black dots on each panel.





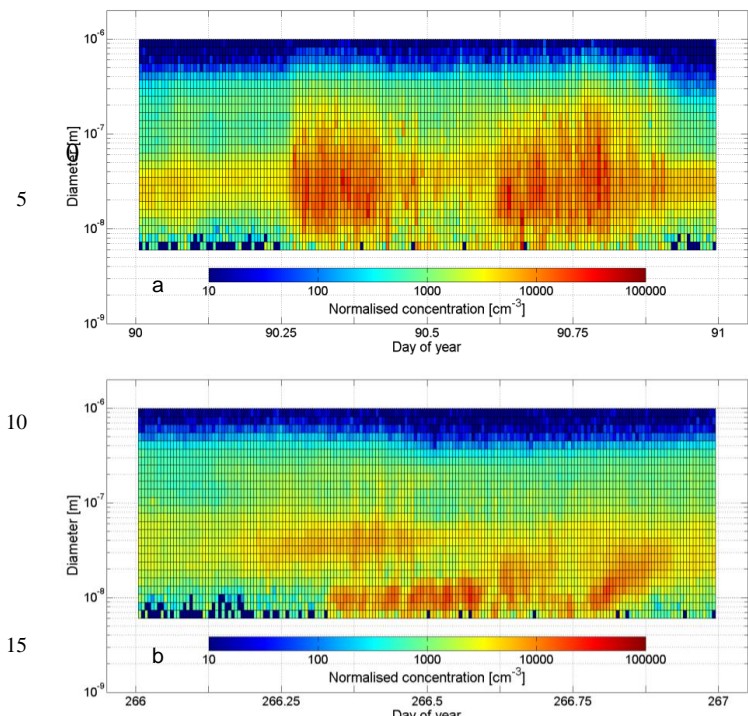

**Figure 11:** Size distribution surface plots showing the typical effects of vehicular road traffic emission sources and local meteorology (a) in the city centre on Tuesday, 31–03–2015; and of extensive grass cutting in neighbourhood (b) of the BpART facility on Tuesday, 23–09–2014.