# Peer review of "Measurement, growth types and shrinkage of newly formed aerosol particles at an urban research platform"

_Atmospheric Chemistry and Physics, 2016_

## Referee Comment (RC1) · Anonymous Referee #1 · 20 Apr 2016

The manuscript discusses several features related to atmospheric new particle formation (NPF) in an urban environment by taking measurements conducted in a research platform located in Budabest, Hungary, as an example. While urban NPF has been studied in a number of papers so far, this manuscript provised several new pieces of insigth into this topic, making the manuscript original enough for publication. The manuscript appears to be scientifically sound, with no major errors in methods or data interpretation. I would recommend accepting this paper for publication after the authors have carefully considered the mostly minor comments outlined below.

My main problem with this paper is that it deals with several topics, all of which are not very tightly related to each other. More specifially, the authors define 6 objectives

[Figure]

(O1. . .O6) for this paper at the end of section 1. I am not saying that some of this material should be left out, but it might help the reader if the authors would organize the list of their objectives somehow. For example, O1 is not really an objective but rather a description of a platform. O2 is a real objective, but closely connected with O1, so O1 and O2 could somehow be tied together. Of the rest, O3 is a method-related objective, while the purely scientific O4-O6 are all related to NPF.

There are a couple of minor issues related to the logic how thing are expressed in the paper. For example, it is a real gas-phase H2SO4 concentration, not its proxy, that causes an atmospheric phenonmenon. The fact that the real concentration was not available, but was estimated using a proxy, is OK but should not be mixed with the true cause-effect relationship. This should be corrected in both abstract and conclusions section. The same problems concerns size distribution surface plots discussed in section 3.5: phenomena like blizzard or emissions from various sources cause chanbes in size distributions, which are then seen in the surface plots, but these phonemena do not affect the surface plots by themselves. Please modify.

The first sentence of the introduction " . . .NPF. . .relevant in urban environments as well" contains information not mentioned in the paper: "as well" gives the impression that NPF may be more important in non-urban environments, but there is nothing in the paper to back up this.

I do not fully understand what is meant in lines 4-5 on page 9. Is limiting the shrinkage something the prevents to shrinkage to take place actively? And how is SO2 exactly related to this, i.e. when one would expect changes in SO2 concentration to affect this process?

I do not understand what "wide variety" refers to in line 4 on page 2.

Finally, the are a few language issues that shoud be corrected:

Articles are missing from several places (e.g by a continuous on page 7, The concentratoin of SO2 on page 9)

Page 2, line 9: systematically overviewed

Page 5, line40: summarized

Page 7, line 29: surface plots

Page 8, lines 22-24: there is something wrong (missing?) in this sentence.

Page 9, line 21: should not, however, be fully. . .

---

## Referee Comment (RC2) · Anonymous Referee #2 · 2 Jun 2016

The manuscript describes and characterizes the aerosol measurement site in central Budapest. A detailed analysis of the measurement systems and the large-scale representativeness of the measurements is given.

Observations of new particle formation (NPF) events are presented, and analysis of several growth and shrinkage episodes is performed. The manuscript provides several hypothesis for the observed particle growth and shrinkage events.

Meteorological conditions and precursor species concentrations are reported as average values calculated over all the observed NPF growth and shrinkage events. It would be interesting to see also comparisons of the precursor and NPF parameters (particle formation and growth/shrinkage rates) on an event-by-event basis. For example, can

the observed shrinkage of the nucleation mode particles be linked to changes in the concentration of precursor vapours? There are not many reports in the literature of particle shrinkage in relation to NPF, so these type of detailed analysis would provide valuable information. I understand if the authors feel that this type of analysis is outside the scope of this paper, but are there any plans for a possible follow-up paper that would focus more on the detailed NPF analysis?

I recommend accepting the manuscript for publication in Atmospheric Chemistry and Physics, after the authors have addressed the comments below.

General comments:

Page 3, lines 38–40: How representative are these gas-phase measurements for BpART (has there been intercomparison measurements between BpART and the air-quality site 1.6 km away)? Could there be larger differences in the concentrations between the two sites when the airmasses do not pass both sites?

Page 7, lines 8–10: What are these properties related to NPF events that varied substantially?

Page 7, lines 37–39: There seems to be a growing nucleation mode present after 9 p.m (Fig 11b). Could this growing NPF also be related to the grass cutting?

Page 8, line 12: "varied" would be a better term than "changed"

Page 8, lines 27–29: What is the typical time that particles were in high-vacuum conditions in the TEM, without evaporating?

Page 8, lines 34–36: What does it mean that the growth and shrinkage phases "were selected by linear approximation"?

Page 9, lines 35–37: In the conclusions the decreasing global radiation and $H_2SO_4$-proxy concentration are highlighted as the main reasons for the observed particle shrinkage events. However, the shrinkage episodes seem to be quite rare (8 out of

178 NPF events), whereas the global radiation and the proxy typically decrease in the afternoons of every day. Could the authors clarify their conclusion? Are the change rates of global radiation and proxy during the shrinkage episodes (reported in Table 3) much higher than on a typical afternoon of an NPF-day without particle shrinkage (for example on the day shown in Fig 8a)?

Page 15, Table 3: What is the reason for using both mean and median values in Table 3? The numbers in the last column are the mean G/S ratios of medians over growth and shrinkage phases, and as such are somewhat different from the ratios of the reported mean values over the growth and shrinkage phases.

Page 15, Table 3: Reporting the $H_2SO_4$ proxy in the units molec/cm$^3$ instead of $\mu$g m$^{-5}$ W s would make it easier to compare the proxy values to those from other sites. I know that the scaling coefficient in Petäjä et al. (2009) is based only on measurements in Hyytiälä, Finland, but it is still widely used in the literature, and would thus enable at least a semi-quantitative comparison of the Budapest proxy values with other sites.

Technical comments:

Page 2, line 9: "systematic" should be "systematically"

Page 2, line 18: "surface based" and "satellite born" should be "surface-based" and "satellite-born"

Page 3, line 20: "ordinary" should be "ordinarily"

Page 3, line 29: "Non-nucleated time intervals" sounds strange. The sentence could be rephrased, e.g. "Only time intervals outside NPF were considered .."

Page 7, line 15: "no" should be "none"

Page 8, lines 5–6: "non-unambiguous" should be "ambiguous"

Page 8, line 31: "pot" should be "plot"

Page 9, line 25: "Chemical and physics situations" sounds strange. Something like "Conditions related to chemical and physical processes in urban atmospheric environments .."

Page 15: The units of $SO_2$ and $O_3$ concentrations should be $\mu$g m$^{-3}$

---

## Author Comment (AC1) · 5 Jun 2016

The authors thank Referee #1 for his/her valuable comments to further improve and clarify the MS. We have considered all recommendations, and made the appropriate alterations. Our specific responses to the comments are as follows.

Comment 1 My main problem with this paper is that it deals with several topics, all of which are not very tightly related to each other. More specifially, the authors define 6 objectives (O1: : :O6) for this paper at the end of section 1. I am not saying that some of this material should be left out, but it might help the reader if the authors would organize the list of their objectives somehow. For example, O1 is not really an objective but rather a description of a platform. O2 is a real objective, but closely

[Figure]

connected with O1, so O1 and O2 could somehow be tied together. Of the rest, O3 is a method-related objective, while the purely scientific O4-O6 are all related to NPF.

Response to Comment 1 We decreased the number and modified the formulation of the objectives of the study. The objective 1 was largely removed from the list, while the rest of it was included into the objective 2, and more emphasis was put on the objectives 4–6, which are considered scientifically more relevant. They were finally reordered as well. See page 2, line 12–16.

Comment 2 There are a couple of minor issues related to the logic how thing are expressed in the paper. For example, it is a real gas-phase $H_2SO_4$ concentration, not its proxy, that causes an atmospheric phenonmenon. The fact that the real concentration was not available, but was estimated using a proxy, is OK but should not be mixed with the true cause-effect relationship. This should be corrected in both abstract and conclusions section. The same problems concerns size distribution surface plots discussed in section 3.5: phenomena like blizzard or emissions from various sources cause chanbes in size distributions, which are then seen in the surface plots, but these phonemena do not affect the surface plots by themselves. Please modify.

Response to Comment 2 The gas-phase $H_2SO_4$ proximity value was found to be proportional to atmospheric $H_2SO_4$ concentration (Petäjä et al., Sulfuric acid and OH concentrations in a boreal forest site, Atmos. Chem. Phys. 9, 7435–7448, 2009; Mikkonen et al., A statistical proxy for sulphuric acid concentration, Atmos. Chem. Phys. 11, 11319–11334, 2011). We utilized the $H_2SO_4$ proxy in the evaluations in a relative way to access the influence of $H_2SO_4$ concentration change on atmospheric properties or processes. The particle number size distribution surface plots express jointly the variation in particle diameter and particle number concentration density in time. The time series of size distributions can be modified by transformation processes (e.g. changes in sink, coagulation or deposition of particles, which often depend on local meteorological and chemical conditions) in the atmosphere as well. Some of these processes can affect the size distributions in general, while some others cause alteration in specific

particle diameter ranges or in certain time intervals. Some sentences in the body, abstract and conclusions of the MS were modified to express our intention more clearly and specifically. See page 1, line 21 and page 10, lines 1–2.

Comment 3 The first sentence of the introduction " : : :NPF: : :relevant in urban environments as well" contains information not mentioned in the paper: "as well" gives the impression that NPF may be more important in non-urban environments, but there is nothing in the paper to back up this.

Response to Comment 3 We modified somewhat the sentence to make it clearer that we are just to refer to historical timeline. NPF and consecutive particle growth processes in the atmosphere were first identified in clean environments. They were shown to be important processes in the global troposphere. NPF has been proved to be common and substantial in polluted environments including large cities as well at a later stage. In addition to its climate effects, NPF in cities can have relevance for public health issues. A reference was also added to back this idea. See page 1, lines 25–26.

Comment 4 I do not fully understand what is meant in lines 4-5 on page 9. Is limiting the shrinkage something the prevents to shrinkage to take place actively? And how is SO2 exactly related to this, i.e. when one would expect changes in SO2 concentration to affect this process?

Response to Comment 4 The sentence was largely reformulated as: "The concentration of SO2 precursor gas was also rather constant during both the growth and shrinkage phases, which suggests that SO2 is usually available in excess in the city (Salma et al., 2011a), and that the processes do not seem to be sensitive to SO2." A reference on ordinary SO2 concentration in Budapest was also added.

Comment 5 I do not understand what "wide variety" refers to in line 4 on page 2.

Response to Comment 5 The expression under question was exchange by "larger number of relevant emission sources of UF particles".

Response to Comment on language issues The whole MS was checked for language and typing errors. The specific examples mentioned by the Referee were, naturally, all corrected.

Imre Salma

---

## Author Comment (AC2) · 5 Jun 2016

The authors thank Referee #2 for his/her detailed and valuable comments to further improve and clarify the MS. We have considered all recommendations, and made the appropriate alterations. We would also like to mention that there was a scatter in some properties of particle shrinkage on an event-by-event basis, which was likely caused by the complex relationships between the particular property and the atmospheric conditions. The fluctuation is expected to be decreased by separating the shrinkage events into groups according to major atmospheric conditions or some other selection principles, and by evaluating them separately for these groups. At the moment, this is unfortunately hindered by their limited number. We plan a dedicated measurement

campaign and evaluation of particle shrinkage on a longer data sets. Our specific responses to the comments are as follows.

Comment 1 Page 3, lines 38–40: How representative are these gas-phase measurements for BpART (has there been inter-comparison measurements between BpART and the air quality site 1.6 km away)? Could there be larger differences in the concentrations between the two sites when the air masses do not pass both sites?

Response to Comment 1 The municipal air quality measurement stations measure criteria air pollutant at several locations in Budapest, and the BpART research platform has not been compared to them, or more specifically to the closest municipal station in upwind prevailing wind direction yet. It could be realized in the future. It is noted, however, that SO2 concentration is ordinary distributed without larger spatial differences within the city (Salma et al., Comprehensive characterisation of atmospheric aerosols in Budapest, Hungary: physicochemical properties of inorganic species, Atmos. Environ. 35, 4367–4378, 2001), and, therefore, its actual value at the research platform is less influenced by air mass directions. In addition, an important advantage of the selected research location at the river Danube (mentioned also in the MS) is that it receives a well-mixed, average air masses from the city centre. The text was extended to include this information, and the reference was added. See page 4, lines 1–2.

Comment 2 Page 7, lines 8–10: What are these properties related to NPF events that varied substantially?

Response to Comment 2 We referred to N_6–25 and gas-phase H2SO4 proxy. They were listed explicitly now.

Comment 3 Page 7, lines 37–39: There seems to be a growing nucleation mode present after 9 p.m (Fig 11b). Could this growing NPF also be related to the grass cutting?

Response to Comment 3 The particle growth observable in Fig. 11b around 21 UTC+2

cannot be considered as NPF event because 1) it does not fulfil the requirements for NPF (Dal Maso et al., Formation and growth of fresh atmospheric aerosols: eight years of aerosol size distribution data from SMEAR II, Hyytiälä, Finland, Boreal Environ. Res. 10, 323–336, 2005), and 2) it also appeared after the sunset. There were no nocturnal NPF events identified at all in Budapest since 2008. The growth was likely related to condensation on existing (Aitken-mode) particles which was caused by changes in local meteorology (e.g. decrease in air temperature and/or air mixing) and atmospheric chemistry. A similar example was discussed earlier (Salma et al., Production, growth and properties of ultrafine atmospheric aerosol particles in an urban environment, Atmos. Chem. Phys. 11, 1339–1353, 2011). A sentence was added to include this argument. See page 8, lines 1–4.

Comment 4 Page 8, lines 27–29: What is the typical time that particles were in high-vacuum conditions in the TEM, without evaporating?

Response to Comment 4 The freshly nucleated particles could withstand the conditions of high vacuum without evaporating for several tens of minutes. It was specified explicitly in the MS together with a reference. See page 8, line 36.

Comment 5 Page 8, lines 34–36: What does it mean that the growth and shrinkage phases "were selected by linear approximation"?

Response to Comment 5 The GR and SR of the nucleation-mode particles were derived as the slope of the fitted linear line of several particle number median diameter (NMD) data in their time evolution plot (Kulmala et al., Measurement of the nucleation of atmospheric aerosol particles, Nat. Protoc. 7, 1651–1667, 2012). These two time intervals were utilised separately as the time intervals for averaging aerosol-related and meteorological properties characteristic for the growth and shrinkage conditions. They are shown in Table 3. The text was modified accordingly. See page 8, lines 40–page 9, line 2.

Comment 6 Page 9, lines 35–37: In the conclusions the decreasing global radiation

and H2SO4- proxy concentration are highlighted as the main reasons for the observed particle shrinkage events. However, the shrinkage episodes seem to be quite rare (8 out of 178 NPF events), whereas the global radiation and the proxy typically decrease in the afternoons of every day. Could the authors clarify their conclusion? Are the change rates of global radiation and proxy during the shrinkage episodes (reported in Table 3) much higher than on a typical afternoon of an NPF-day without particle shrinkage (for example on the day shown in Fig 8a)?

Response to Comment 6 The explanation was extended by condensable vapours in general since the mean contribution of H2SO4 condensation to the particle GR was estimated to be 12.3% in Budapest (Salma et al., Regional effect on urban atmospheric nucleation, Atmos. Chem. Phys. Discuss. doi:10.5194/acp-2016-115, in review, 2016). Oxidation products of VOC are expected to play a more important role in the growth of relatively larger particles (with a diameter above ca. 20 nm). The shrinkage is likely caused by a joint effect of multiple factors including substantially decreased concentration of condensable vapours and GRad. Sudden changes in GRad could inhibit NPF or reverse the ongoing particle growth (Baranizadeh et al., The effect of cloudiness on new-particle formation: investigation of radiation levels, Boreal Environ. Res. 19, 343–354, 2014.) See also the arguments in the introductory part of this Authors' response. The text of the MS was modified to express these details in more detail. See page 10, lines 2–4.

Comment 7 Page 15, Table 3: What is the reason for using both mean and median values in Table 3? The numbers in the last column are the mean G/S ratios of medians over growth and shrinkage phases, and as such are somewhat different from the ratios of the reported mean values over the growth and shrinkage phases.

Response to Comment 7 The atmospheric properties in Table 3 were expressed by their mean values. The ratios of these properties for the growth and shrinkage phases were, however, calculated by using their median values since the latter values are less sensitive to large fluctuations than the means, and therefore, they seem more

advantageous for reliability and propagation of errors.

Comment 8 Page 15, Table 3: Reporting the H2SO4 proxy in the units molec/cm3 instead of $\mu$g m$-5$ W s would make it easier to compare the proxy values to those from other sites. I know that the scaling coefficient in Petäjä et al. (2009) is based only on measurements in Hyytiälä, Finland, but it is still widely used in the literature, and would thus enable at least a semi-quantitative comparison of the Budapest proxy values with other sites.

Response to Comment 8 The gas-phase H2SO4 proxy value was calculated as [SO2]$\times$GRad/CS for intensities >10 W m$^-$2. We would prefer using it without the scaling factor k between the proxy value and H2SO4 concentration since the scaling factor was derived specifically for a remote boreal site as an empirical relationship: k=1.4$\times$10$^-$7$\times$ GRad$^-$0.70 (Petäjä et al., Sulfuric acid and OH concentrations in a boreal forest site, Atmos. Chem. Phys. 9, 7435–7448, 2009). Urban areas can/are expected to differ from the remote regions (Mikkonen et al., A statistical proxy for sulphuric acid concentration, Atmos. Chem. Phys. 11, 11319–11334, 2011), and the GRad involved in the scaling factor can distort the relationships investigated. As far as the comparison of our data to other sites is concerned, we referred now to a recent paper (Salma et al., Regional effect on urban atmospheric nucleation, Atmos. Chem. Phys. Discuss. doi:10.5194/acp-2016-115, in review, 2016) in which the scaling factor was adopted for the Budapest H2SO4 data.

Response to Technical comments All technical comments were accepted and adopted.